# "We need our own clinics": Adolescents' living with HIV recommendations for a responsive health system

**Nataly Woollett**[1]*, **Shenaaz Pahad**[2], **Vivian Black**[3]

**1** Wits School of Public Health, Faculty of Health Sciences, University of Witwatersrand, Johannesburg, South Africa, **2** Wits Reproductive Health & HIV Institute (Wits RHI), Johannesburg, South Africa, **3** Department of Clinical Microbiology and Infectious Disease, Wits School of Pathology, Faculty of Health Sciences, University of Witwatersrand, Johannesburg, South Africa

* woollettn@gmail.com

**Data Availability Statement:** Please note there are special circumstances regarding this research. A court order was needed as a result of a change in legislation in South Africa and on account of participants not having legal guardians, i.e. it is not

## Abstract

Adolescents living with HIV comprise a significant patient population in sub Saharan Africa but are poorly retained in care with consequent increased mortality and morbidity. We conducted in-depth interviews with 25 adolescents living with HIV engaged in care from five clinics in Johannesburg regarding their recommendations for the healthcare system. Findings included advocating for adolescent clinics, recognizing the importance of clinic-based support groups, valuing the influence lay counselors have in providing healthcare to adolescents, improving widespread education of vertical HIV transmission and meaningfully linking clinics to the community. Our study offers guidance to the differentiated care model recommended for adolescent treatment highlighting that a positive youth development approach and use of lay and peer counselors may act as cornerstones of this model. Serving the mental health needs of adolescents living with HIV in a responsive manner may strengthen their use of the system and elevate it to a source of resilience.

## Introduction

Adolescents living with HIV comprise nearly 2 million in sub-Saharan Africa (SSA) [1], a region disproportionately affected by the virus. There is mounting evidence indicating adolescent living with HIV are progressively poorly retained in care, have low rates of adherence and viral suppression, with increased mortality and morbidity [1–3]. Vertically infected children are reaching adolescence in large numbers and incident HIV infection is high in this age group, particularly in adolescent girls [4, 5]. These rates are concentrated in SSA where health systems tend to be overburdened and poorly resourced, especially the case in South Africa [6] and where adolescents comprise a sizeable proportion of the population [7]. The health system in Johannesburg is overwhelmed with large numbers of patients, often in excess of healthcare providers; leading to rapid expansion of a range of paraprofessionals being employed to address services delivery gaps [8, 9]. Healthcare systems in SSA will be managing adolescent HIV for decades to come [10], a reality calling for intentional investment to curb onward

customary for children whose parents have died and who go on to live with other caregivers, for those caregivers to approach the state to become 'legal guardians'. As such, I had to approach the high court in Johannesburg to give consent to recruit participants. The court order does not give me consent to share data. For researchers interested in the data set, please contact Professor CB Penny, Chairperson of the Human Research Ethics Committee (Medical) at the University of Witwatersrand, on telephone no. +2711 717 2301, or by e-mail at Clement.Penny@wits.ac.za. Please reference ethics approval number M130258.

**Funding:** The author(s) received no specific funding for this work.

**Competing interests:** The authors have declared that no competing interests exist.

transmission and viral resistance, and improve maintenance and care for this vulnerable group [11].

Recent systematic reviews have endeavored to unearth the reasons contributing to the poor health outcomes of adolescents living with HIV. Reasons identified include stigma, perceived lack of confidentiality at clinics, poor treatment by healthcare providers, treatment fatigue, ineffective transitioning from pediatric to adult clinics, poor disclosure, and lack of adolescent-friendly services [12–14]. At present there are limited best practices to mitigate these problems [15, 16]. There are currently no national guidelines on how to transition adolescents when they reach adulthood at 18 years or before if they are stable on treatment [17]. There is recognition that adolescence is a challenging developmental period, marked by rapid physiological, psychological, familial and social change [18]. These challenges are compounded for adolescents living with HIV, who tend to have a 'stacking of odds against them' in terms of psychological, social and health needs [19, 20]. However, there is also a dearth of evidence of what improves outcomes, particularly under the 'treat all' strategy, acknowledging that more comprehensive healthcare is required for this group and further commitment from all who serve them [21]. Differentiated models of care have been proposed to manage adolescent HIV in public health systems, but substantiation on how to implement these well remains imprecise [22, 23]. Specifically, with the scant confirmation from adolescents themselves on what works for them in terms of their HIV care and treatment [24].

Our study sought to address this research gap by asking adolescents living with HIV engaged in care for many years about their recommendations for a more responsive healthcare system in adolescence.

## Methods

### Participants and procedures

This study (n = 25) nested qualitative research in an observational study (n = 343) conducted in five HIV clinics serving adolescents in Johannesburg, South Africa. Three of the clinics were hospital based; one was a community health center and one a primary healthcare clinic. All offered routine care except one hospital based 'flagship' clinic that offered specialized pediatric ART services. All facilities had an 'adolescent clinic', generally a day when adolescent patients were scheduled for care, except the flagship clinic that offered adolescent care every day. These clinics typically saw patients aged 13-22yrs for treatment. There was a push to refer stable patients on ART to more general adult clinics to manage clinic congestion and many of these were generally referred around age 12-13yrs. All clinics were located in urban settings typified by poverty, violence, and poor infrastructure (6). Data was collected over a period of eight months.

Clinic-based research counselors familiar to the adolescents living with HIV purposively recruited them based on consent and willingness to engage in in-depth interviews. Research counselors were clinic based lay counselors trained in pediatric HIV and advanced counseling, and volunteered to participate in the research, engaging in principles of ethical research with minors. They received an additional five-day training for this study as well as weekly debriefing and supervision throughout the course of data collection.

All participants reported experience of being disclosed to regarding their HIV-status, were on treatment for HIV infection, were currently retained in care, had been accessing treatment for a number of years, and participated in a clinic-based support group. Most had histories of varying levels of adherence to treatment as well as retention. Twenty-five adolescents were recruited equally from all five facilities. The first five participants that met inclusion criteria at each of the five sites were invited to participate in the qualitative interviews. No participants refused and we felt we reached saturation with 25 participants in total. They ranged in age

from 13–19 years (mean 16 years; 15 female). Participants were infected at birth or early childhood and had lived with HIV as long as they remembered. Participants were interviewed for approximately 60 minutes by (author) using a semi-structured interview guide at their clinic visit. The interviews were audio recorded with participant consent and conducted in English as all participants spoke English fluently.

Participants were given a pocket-sized card with active referrals for services in their area including counseling, legal and crisis resources. Upon completion of the interview, they received reimbursement for travel; a snack; and a gift voucher for R50 (US $3) at a local clothing retailer. These items were endorsed and recommended by the adolescent community advisory board (ACAB) instituted for the research. The ACAB gave advice on methods and were also engaged with the findings of the study.

## Characteristics of adolescent clinics

Adolescent clinics were implemented by scheduling all 13–19 year old patients on the same day, creating a critical mass of young people to engage with from all five clinics. All staff implementing the clinics were trained in child and adolescent development and the needs of HIV infected young people, i.e. disclosure, adherence, retention in care and supportive down referral. Values clarification exercises were undertaken to ensure a rights-based approach was understood.

Open clinic support groups were offered and any adolescent was welcome to join after their visit with the doctor/nurse. Groups changed weekly based on adolescent visits to the clinic and most groups comprised 10–40 participants depending on the clinic. Not all adolescents attending these clinics were aware of their HIV status or had been disclosed to. Consequently support groups covered topics such as taking medicine in correct doses at specific times, the importance of adherence and the need to be adherent for the medication to work, managing school difficulties and other similar issues raised by the adolescents. No overt discussion of HIV infection was provided. In addition, caregiver support groups were offered at the clinic with an emphasis on treatment literacy and supporting caregivers to disclose HIV status to children and adolescents in their care.

Closed support groups were hosted on Saturdays and during school holidays. These groups tended to have consistent participation over time and comprised approximately 20 adolescents per group. Inclusion required that adolescents knew their HIV status and had disclosed their status to at least one other person, making it safe to speak openly about HIV and ART. The content of these groups ranged from understanding of medication regimes and ART adherence requirements for viral suppression, to adolescents living with HIV specific concerns such as how to manage adherence on weekends and holidays amongst friends or family when one had not disclosed, what to do when one missed a dose, treatment fatigue and medication vacations/ weekends off, self management skills, dealing with disclosure, discrimination and internalized stigma. Regular adolescent issues such as sexual reproductive health matters, adolescent pregnancy, romantic relationships and gender equality, managing peer pressure, mental health problems (particularly trauma, depression, bereavement and suicide) and caregiver communication challenges were also explored.

The closed groups subscribed to a rights based, participatory practice [25] and positive youth development model [26]. Positive youth development programs included actively promoting personal agency and responsibility, using respectful means to engage young people and focus on their strengths and competencies to help them overcome some of their adversities and risks [26]. Instead of viewing adolescents as patients with 'problems to be managed', the view is rather that adolescents are resources to be developed and have the means to critically engage in solving their own difficulties. Adolescents defined what they wanted to cover in

the groups and how. Frequently non-verbal means of working were utilized such as art and play therapy. Experts were invited to present on topics such as yoga and mindfulness practices, self-defense skills, nutrition, and job readiness. These groups were co-led by psychologists, social workers and lay counselors and lasted approximately 3–4 hours. All facilitators received monthly supervision and debriefing from experienced mental health clinicians. Lunch and transportation costs to attend the groups were provided.

## Ethics

The South African National Health Act states that 'for health research with minors (<18-year olds), consent from a legal parent or guardian for research with children must be obtained' (Section 71, 2012). It was speculated that the proposed study participants may not have legal guardians from whom to obtain consent. Resultantly, the ethics committee of the University of the Witwatersrand advised seeking permission from the court to enroll participants in the study. A court order was granted that led to full ethics clearance and permission to interview participants without parental or guardian consent (M130258). Adolescents gave verbal assent and written consent to participate in this study. All participation was sought on the basis of good clinical practice guidelines and protective procedures were built in to support adolescents in need of additional assistance. Permission was also granted from Gauteng Provincial Department of Health, Johannesburg District Department of Health, and the research committees of individual facilities where participants were recruited from.

## Data analysis

All interviews were transcribed verbatim in English. All identifying information was removed and transcripts were saved by a coded name. Data were managed in QSR NVivo 10, constructing an analytical framework of broad codes by generating a 'start list' of potential themes and building upon the research questions. Each broad code, or wide thematic basket of concepts [27], was applied to every transcript and 'fine codes' were expounded using an inductive method deriving meaning from the data itself as opposed to imposing pre-formed impressions [28]. To ensure intercoder consensus, fine codes were developed by three researchers experienced in qualitative data analysis by printing out a full set of excerpts (from each data set) associated with each code for each transcript and identifying sub-themes arising from the data. Two researchers applied the thematic codes to every transcript. Each week during the double coding, the findings were critiqued and discussed within the group to guarantee research results. During analysis, we found that data reached a point of saturation, which suggests the sample size was sufficient.

## Results

Please see Table 1 for a description of the sample.

### 1. Advocating for adolescent clinics

**1.1 Being with peers less threatening than being in adult clinics.** Many stable adolescent patients had been down referred to adult clinics and soon returned to the adolescent clinics, Kagiso and Bafense below remembered this experience. Findings indicated that many adolescents did not have the agency and confidence to navigate adult settings on their own and felt it would be more comfortable to receive healthcare within their own peer group.

*Why send a kid who is only starting to learn about life, his situation in life, the purpose for their life to an adult clinic? How are they going to speak to an adult who has already been*

**Table 1. Description of the sample.**

| Name (pseudonym) | F/M | Age | Know how became infected? | Orphan status | Living arrangement |
|---|---|---|---|---|---|
| Jabulile | F | 15 | No | Double | Grandmother |
| Kagiso | M | 17 | Yes | Double | Children's home |
| Banele | M | 18 | No | Double | Grandmother |
| Ayanda | F | 16 | No | Double | Foster home |
| Nobuhle | F | 19 | Yes | Double | Aunt |
| Buhle | F | 15 | No | Single | Grandmother (bio father alive no relationship) |
| Thabisile | F | 17 | Yes | Double | Grandmother |
| Sindiswa | F | 16 | No | Single | Bio father and stepmother |
| Phindile | F | 16 | No | Double | Aunt |
| Zodwa | F | 18 | Yes | Non | Bio mother (bio father alive no relationship) |
| Lesedi | M | 13 | Yes | Double | Great grandmother |
| Tebogo | F | 18 | Yes | Single | Bio mother |
| Tshepo | M | 18 | Yes | Non | Bio mother (bio father alive no relationship) |
| Sithembile | M | 18 | No | Double | Grandmother |
| Nomphumelelo | F | 17 | No | Double | Grandmother |
| Ntokoza | M | 13 | No | Double | Grandmother |
| Palesa | F | 14 | Yes | Single | Grandmother (bio mother alive poor relationship) |
| Lerato | F | 18 | Yes | Non | Bio mother (bio father alive poor relationship) |
| Mahlatsi | M | 17 | No | Single | Bio mother |
| Bafense | M | 16 | Yes | Double | Children's home |
| Ntombi | F | 14 | No | Single | Bio mother |
| Amahle | F | 14 | Yes | Double | Aunt |
| Mati | F | 18 | Yes | Double | Grandmother |
| Bongani | M | 18 | No | Single | Bio father |
| Siphiwe | M | 18 | No | Non | Bio mother (bio father alive no relationship) |

Note all names attached to quotes are fictional to protect confidentiality.

*through their life issues about a kid who's only growing up? It's best to put him in a clinic where he's in his own age group. Everything is understandable to kids to their age group so I don't understand the point in sending a kid to a clinic where it only accommodates older people. It doesn't make sense–Kagiso, male 17yrs*

Some reported being disrespected in adult clinics.

*Because the nurses in the children's clinic are more polite than the ones in the other clinics and so they make sure they enjoy coming to the clinic. . .no one is going to shout at me or do anything, unlike the other clinics–Bafense, male, 16yrs*

**1.2. Regular health system increases discrimination towards adolescents living with HIV.** It appeared that within the adolescent clinic, their HIV status may seem anonymized but at adult clinics, where there are fewer adolescents, they would be more conspicuous and their status more apparent.

*It's hard for them because they're sitting with adults. Some of the adults are in their communities, they're going to start talking about what happened to this kid and all those things–Ayanda, female, 16yrs*

Similarly there seemed to be a suggestion that many adults, including nurses, would not understand vertical transmission and might discriminate against adolescents living with HIV.

*The nurses there don't really understand. They don't come to terms with the kids who is on ART, they'd simply think that the kid was reckless when that wasn't the main reason behind the infection–Sithembile, male, 18yrs*

As opposed to adolescents all being treated for HIV, participants highlighted that most adult clinics don't integrate care and in that way the system facilitated inadvertent HIV disclosure and discrimination.

*Other clinics they differ with things. Say TB here, HIV here, sugar [diabetes] here, that's not what you want to do. . . because of this, people they will see, people are not just like them and we'll start to hate each other and maybe your friend comes in and see you in the HIV line, what he will say, he'll go and spread it and they will laugh you all over and hey, talking silly things about you–Mahlatsi, male, 17yrs*

**1.3. Appeal of adolescent-friendly attitude.** A strong theme emerged highlighting that adolescent clinics tend to have staff that understand and respond to adolescent health and developmental needs enhancing their access to treatment and retention in care.

*The people here are like, they have a mind of a kid which is very good. They know what a kid is thinking, they're very supportive. . .they know what a kid has to go through to fight this kind of thing–Kagiso, male, 17yrs*

All participants appreciated consistent, supportive and non-judgemental staff working within a culture of care towards young people.

*They never, actually they always encourage me to take my medicine because they know that, they say that I must take care of myself so I must always take my medicine. They were never like talking bad things to me, they always telling me, advising me, if you keep on taking your medicine well, you can grow well and you can be as healthy as a person that doesn't have this disease–Sithembile, male, 18yrs*

There was also an appreciation of having a long-term relationship with staff and being known over time with counseling being offered on a regular basis.

*Because they are always there when we needed them and they give us information, they teach us everything we want to know in life, they know me and they know my family–Nobuhle, female, 19yrs*

Part of the appropriate care adolescents received was being told the facts about their disease and its treatment.

*Because they tell you they don't like hide secrets because she is going to like [upset]. . .they tell you straight you are like this and that's what you must do to handle it–Phindile, female, 16yrs*

**1.4. Attributes of convenient healthcare for adolescents.** There was support for adolescent clinics being scheduled after school (most regular clinics work on a 'first come first served basis').

*You could set a schedule for them by timetable, OK, maybe attend when they come back from school so they don't miss out on schooling–Banele, male, 18yrs*

In busy healthcare environments, an effective means of creating an adolescent clinic is to schedule all patients who are adolescents on the same day.

*Yes, that's why I choose to go on Fridays because there it's just young people–Bongani, male, 18yrs*

Although there was some support for caregiver involvement. . .

*I think the parents is supposed to be getting counseling with the child, going together and understanding, learning more about HIV so that they get full information and so that they understand and when they understand then they disclose their status because some parents they say they fear that the child will kill themselves–Lerato, female, 18yrs*

There was also recognition that adolescents need privacy with their healthcare professionals to meet their needs with confidentiality.

*Okay adults can accompany the kids, but the adults, as far as they go is the door, what happens inside is all about the kids for the kids by the kids. Adults shouldn't be involved in the life of an adolescent kid who's on ARVs–Kagiso, male, 17yrs*

## 2. Value of support groups

Findings revealed robust endorsement of clinic-based support groups for adolescents.

**2.1 Open disclosure was liberating.** The idea of 'being free to talk' underscored the gravity of secrecy in the lives of many vertically infected adolescents living with HIV and highlighted how few opportunities these adolescents had to practice safe ways of disclosure.

*No, because all of us we are HIV positive like teenagers. Even if we are as a group, ja [yes] like we feel free because we are all the same. Like even with status, there's no one with negative status–Amahle, female, 14yrs*

*Some of them never said anything about their status. . .that's the only opportunity because you are all at the same stage, we are all HIV positive, so you can be free sometimes to talk– Mati, female, 18yrs*

*At the end of the day you won't have secrets with your friends here–Phindile, female, 16yrs*

**2.2. Group-facilitated support, shared problem solving and the realization that adolescents were not alone.** Young people recognized the value of peers similarly placed in life being able to give authentic support around shared issues.

*That's where people at your age, who are also in your situation, so those people teach you what they know and you learn from them. So it's helpful–Tshepo, male, 18yrs*

*Because there are other children who are HIV positive and just like can help me understand what is going on in the support group–Ntombi, female, 14yrs*

Many participants identified the group as a means of finding solutions to personal problems.

*For me, I think it's a good idea to have support group and talk about how we feel about even living with HIV, everyone of us must come with the ideas and then we discuss the problems that we're having. . .so for us to have support groups it can be easy for us to discuss or to talk about how we feel about this thing, how can we do, to deal with this disease–Phindile, female, 16yrs*

Many vertically infected adolescents may feel isolated and alone, not realizing there are other young people infected as they were.

*Sometimes it makes me feel that I'm not alone and all that, so ja [yes], it didn't only happen to me and all that–Bongani, male, 18yrs*

*It was like the way people shared their problems, it made me feel like I'm not the only one who is having the same problem as I do–Lerato, female, 18yrs*

Support groups also facilitated mental health gains such as hope, optimism and relief of burdensome emotions.

*They will have to support us in what we are doing because we will help this virus to get down and down because if they are not supporting us, this virus will busy getting up and higher because we are losing a lot of youth because they are scared to be HIV, but as we are, we are not scared cause they mustn't lose hope, there is hope out there, we can help them handle person with pain–Sithembile, male, 18yrs*

*Discuss everything that you want to discuss about you take out all the pain inside you, there by the group–Tebogo, female, 18yrs*

**2.3. Benefits of adolescent-friendly ways of expressing oneself.** Young people who attended the group particularly valued the use of non-verbal ways of expressing feelings and ideas.

*We would get to speak whatever's on our mind and even express our feelings and some with drawing and so on, so you get to relieve your anger and all that–Zodwa, female, 18yrs*

**2.4. Learning more about HIV and sexual reproductive health issues.** Although adolescents had generally been taking treatment for a long time, there was still need to educate about their disease and its impact. A support group structure was effective in this outcome.

*Many children don't understand their treatment because their parents or their caregivers aren't telling them about it, so the only place they get good information is at the clinic–Sindiswa, female, 16yrs*

Very few adolescents had an accurate understanding of vertical transmission or prevention methods such as prevention of mother-to-child transmission (PMTCT), reflecting that they were not being well informed around sexual reproductive health (SRH) and future parenting options. In fact, the interviewer explained PMTCT at length to all participants as there was not comprehensive understanding of this prevention method in the sample.

*Interviewer: So do you know how you contracted HIV?*

*Interviewee: Not really, but by him [the doctor], he thinks maybe I got it at birth, but I think if I got it at birth maybe I would have been dead by now, I'm not sure, so I don't know how I got it–Bongani, male, 18yr*

*Interviewee: That used to freak me out, just thinking of the fact that if I would just get pregnant, what if I also do the same thing to my own child? Maybe my own child wouldn't forgive me, would always be angry like I am to my mom.*

*Interviewer: What does it feel like to know you have these options [PMTCT] and that the chances of you transmitting the virus are very small?*

*Interviewee: It's kind of relieving–Thabisile, female, 17yrs*

## 3. Significance of inclusion of lay counselors into multidisciplinary clinic teams

Many participants spoke about the important relationships they had with lay counselors at the clinic and the value this cadre of worker had in providing education and support to patients.

*Interviewee: Well, I can be honest with them all, you know, because the doctor, I have to go to the doctor first and the doctor will also take me to the counselor.*

*Interviewer: But it's easier to talk to the counselor or is it the same?*

*Interviewee: It's not the same because the doctor just advises you not to do it, just to take your medication. So the counselors just ask you questions, again and again, you know, you talk–Mati, female, 18yrs*

Unmet sexual reproductive healthcare needs were typically met by lay counselors.

*Interviewer: So if you need help with like guy stuff, like I don't know, how to have safe sex, I don't know what you might talk to a guy about, do you know what I mean? But if you did, who do you talk to about that stuff?*

*Interviewee: At home there is no one, but I'll come and talk to [male counselor], because he told me that I can come and talk to him with anything.*

*Interviewer: Ah. . .so you and he are tight [close]?*

*Interviewee: Yes.*

*Interviewer: He's a good counselor.*

*Interviewee: Very–Siphiwe, male, 18yrs*

Participants seemed to rely heavily on counselors for information about HIV and their treatment, but also sought guidance and emotional support from them.

*We need to put more counselors in clinics to support us, maybe telling us what is going on about our lives, what should we do if there is time that's come difficult maybe–Nomphume-lelo, female, 17yrs*

Counselors appeared to be perceived as more trustworthy than other personnel and more likely to maintain confidentiality.

*Interviewee: Because if you have counselor you are open to say things with her, others didn't have. Maybe I told you [healthcare professional] my status, OK, they are writing, then others come. . .you say ah, 'you see that lady is HIV positive' and that's not nice at all.*

*Interviewer: So you feel like counselors are more confidential and they don't necessarily just tell people other people's status in the clinic?*

*Interviewer: Yes, others tell but counselors don't–Tebogo, female, 18yrs*

## 4. Employment opportunities for HIV infected youth within the healthcare system

A theme around young people themselves being active in the care of other HIV infected youth within the health system became evident.

*Well, kids understand kids better. Parents are too controlling over kids and how they live their lives. Obviously parents want their kids to live their lives a certain way because they couldn't live their life the way that they wanted to so obviously so, they want their kids to live their life so just too much stress for a kid at the time. This side is the parent, this side is the medication. There's too much–Kagiso, male, 17yrs*

Some participants identified their ability to add value and strengthen the health system's treatment of infected youth by being employed within it.

*Maybe give people the work, you see, like I'm having HIV, then they offer me a job to go maybe in KZN, I could do that, I don't mind. Maybe talking to a person like just the way we do, encouraging, teaching people how to take their medicine in order to have a better living–Banele, male, 18yrs*

*If I was talking to the minister of health, I would tell him that he must hire people like me that can talk to the children and tell them why they are taking their treatment–Ayanda, female, 16yrs*

## 5. Need for widespread education regarding vertical infection

There appeared to be a persuasive theme advocating for widespread education to the public around vertical HIV infection. Participants shared that there isn't a common understanding of vertical transmission and most assume young people are infected through sexual transmission. This limits their motivation to disclose their status to others.

*If I disclose to my girlfriend she will feel sad and she will think that I slept with other girls. . .we need to advertise that HIV it's not what we get only from sleeping with other people–Lesedi, male, 13yrs*

*I think even negative people must be told about these things [vertical infection], must be told, so that they can understand what is going on–Tshepo, male, 18yrs*

### 6. Meaningfully linking clinic to community

An interesting theme became clear around linking the clinic to community activities, strengthening the adolescent community's ties to the clinic. One participant reflected on a soccer tournament that was hosted in the community and comprised teams from various clinic-based support groups for HIV infected youth.

*We went for a soccer tournament and then we had so much fun. Even girls were playing soccer and all that so you get to enjoy you know what even though you're an HIV person you get to enjoy a lot of things in life–Thabisile, female, 17yrs*

There was also support for school-based testing from participants.

*Yes, even if the schools, I think there must maybe be doctors to come to the school and check children if they are HIV positive or not. . . Because other parents they are scared to come to the hospitals yes, they think that if I have HIV I will die, what if my child finds out, what if my child does this. I think they should try it–Palesa, female, 14yrs*

## Discussion

Our study highlighted adolescent recommendations for a more effective health system that included advocating for adolescent clinics, recognizing the importance of clinic-based support groups, valuing the influence lay counselors have in providing healthcare to adolescents, improving widespread education of vertical HIV transmission and meaningfully linking the clinic to the community.

Mental health needs of children and adolescents globally do not get the attention and resources required, but are severely underserved in low and middle-income countries (LMICs) [29, 30]. Adolescents living with HIV tend to have mental health challenges that impact negatively on adherence and retention in care [31]. Health systems that appreciate this reality may be more responsive to it. There is growing evidence that clinic-based support groups help retain adolescents living with HIV in care and if managed well, can be taken to scale [14, 32]. Groupwork is a powerful method in reaching young people who tend to enjoy the company of their peers, and in resource-constrained contexts, it is a cost-effective means of intervention. The social component of group treatment (i.e. cohesion, interpersonal learning etc.) is a dominant mechanism through which change occurs [33], facilitating psychoeducational and mental health benefits. Support groups are valuable for education, i.e. facilitating understanding through experiential learning rather than simply providing information. Those that intentionally address some of the mental health difficulties of adolescents living with HIV and that are facilitated by staff grounded in understanding adolescent development and mental health, could mitigate risk, improve health system use and treatment adherence.

Keeping secrets is onerous and burdensome through development, a finding articulated in this study. Young people welcome honest communication with adults who serve them [34]. Social support is particularly appreciated for those struggling through development with prolonged mental health problems, depression and complicated grief being particularly noteworthy for vertically infected youth who are frequently orphaned [2]. Depression and complicated grief are associated with young people's perceptions and poor understanding of death in epidemic regions, and need to be addressed in psychosocial programming [35]. Acceptance and belonging cannot be underestimated in terms of adolescent identity formation and development especially with vertically infected young people who commonly experience feelings of isolation and rejection [36]. This reality is underscored in our study with participants placing strong emphasis on adolescent clinics and clinic-based support groups for similarly placed adolescents living with HIV.

Many vertically infected children and adolescents live in communities characterized by poverty and exposure to violence [37] as was the case in our sample [31]. Women living with HIV tend to experience higher rates of intimate partner violence [38] and adolescents exposed to this kind of violence show not only poorer mental health outcomes [39], but also have poor virological and immunological outcomes [40] and poorer adherence [41]. In addition, orphaned adolescents are at risk for heightened exposure to physical, emotional and sexual violence [42]. Cumulative exposure to adversity such as violence, abuse and neglect is associated with increased risky sexual behavior amongst vertically infected adolescents in South Africa, leading to higher probable rates of secondary transmission and potential reinfection and resistance to ART [43]. Very few adolescents living with HIV report disclosing their HIV status to their sexual partner [44], possibly on account of the reality of discrimination from peers [45]. Disclosure protects against mental health risks [2] and as our participants describe, is liberating. Having opportunities to practice disclosure is vital for this group and is encouraged in support groups and adolescent clinics.

Adolescents need more reliable and comprehensive information regarding vertical transmission. Our participants had both a lack of understanding of PMTCT and future reproductive health options, as well as unreliable comprehension of how they contracted HIV and pathways to vertical transmission. These findings are similar to others [46] but highlight that many providers may make assumptions that vertically infected adolescents understand vertical transmission when that may not be the case [47]. Having opportunities to fully and repeatedly explain these mechanisms is important and as our study reveals, is strengthened by strong interpersonal relationship with healthcare providers.

Adolescents living with HIV are sexually active and have intentions of parenthood. However, evidence indicates that female adolescents living with HIV rarely discuss pregnancy intentions with family or providers, suggesting a potential fear of being judged for wanting to conceive when living with HIV [48]. Indeed, 16% of all adolescents in South Africa experience pregnancy, many of whom will be or are HIV infected [7]. HIV infected pregnant adolescents show much higher rates of mother-to-child transmission [49] perhaps on account of not fully understanding vertical transmission or the strength of prevention methods such as PMTCT. Another potential contributor is not feeling comfortable in antenatal clinics where adolescents make up a minority of patients. Health providers need to explain modes of transmission and educate adolescents living with HIV regarding family planning and contraception, including routinely screening for fertility intentions and strengthening access to safer conception assistance. Safer conception services, that support the reproductive goals of women living with HIV and their partners, can minimize the risks of HIV transmission through pregnancy, can enhance ART adherence and promote limiting pregnancy attempts until viral suppression can be achieved and maintained [50]. In addition, vertically infected pregnant adolescents are

likely to experience poorer adherence on account of depression in the peripartum period [51] indicating that mental health screening and support in this population is prudent.

Effective programming to reduce sexual risk behavior of adolescents living with HIV is essential. Unfortunately, topics such as sexual and reproductive health are taboo and are not easily spoken about in families, thus healthcare environments are positioned to fill this gap with factual and developmentally appropriate information [52]. In addition, evidence shows the manner in which young HIV infected patients want to be engaged in SRH matters is similar to their HIV management, i.e. adolescent-friendly and convenient with improved service quality from clinic staff [53]. Our study also highlights the benefits of investing in male providers to address SRH education and treatment.

Lack of understanding of vertical infection and discrimination towards adolescents living with HIV in society, especially by adults, leads to young people's distrust of adults and particularly health system personnel [54]. However, in our study, counselors appeared to be trusted to maintain confidentiality, take time to explain things, provided honest and understandable information, and were a constant presence in the clinic that were highly prized. The lay counselors conducted many individual 'adherence counseling' sessions with patients but were central in delivering the psychosocial support groups held at the clinic.

Lack of skilled mental health professionals available to intervene and the need to utilize and upskill lay counselors to meet the demand ethically with targeted training and consistent expert supervision [55, 56] is now vital to public health systems. There is expanding proof that lay counselors are able to provide skillful and effective task-sharing interventions locally [57] and our study provides additional support for this.

HIV infected youth have the ability to educate peers, both those uninfected as well as newly infected, as they understand healthcare utilization, the challenges of care and treatment over time, as well as youth difficulties. Adolescent peers have influence over each other, especially those that are marginalized in similar ways. Friendships with peers moderate personal adversity [58], and HIV infected youth with other infected youth in their social networks report being able to accept their diagnosis more easily [59]. Care models that capitalize on this may have influence in retaining adolescents in treatment and impact HIV related stigma in beneficial ways [60]. As such, programs that work with adolescents living with HIV may profit from peer interventions at clinic level in the shape of peer led support groups, peer treatment navigators/counselors for youth (supervised by trained facilitators), and inclusion of young adolescents living with HIV in multidisciplinary teams. Peer educators or counselors located in adult clinics may also help with the transition of stable adolescents from pediatric to adult clinics and keep them retained in care. They may increase adolescent access to testing and timely initiation on treatment. In tandem, these young people would be gaining constructive workplace skills that can be capitalized on as they age into the workforce.

Linking clinic to community through activities such as soccer tournaments and informative productions drives home the positive aspects of health and depathologizes HIV; it was a significant recommendation from this study. Community-based adherence clubs that have shown positive outcomes for adult patients [61] may be equally effective for adolescents, especially if there is a support group component to implementation and if the groups are adolescent only [32].

A positive youth development style to service delivery of adolescents living with HIV is recommended to foster adolescents' growing autonomy, actively including them in decision making, highlighting their rights and also their responsibilities through positive, respectful and mutually beneficial relationships with service providers [26]. Offering opportunities for adolescents living with HIV to exercise their agency leads to engagement and positive health benefits including accessing their resilience [62]. It also adds another resource available to young people: a responsive and supportive healthcare system. An adolescent's ability to cope is

influenced by the degree to which they are able to connect with their community and negotiate assistance from it thus enabling environments that include their participation lead to optimal coping. As such, the healthcare environment could be actively employed as a coping-enabling social environment for adolescent patients, facilitating opportunities for resilience, and promoting participation as a tactic to cope with adversity [63]. Indeed, recent resilience studies in South Africa have shown perceptions of care are associated with improved help-seeking behaviors from adolescents, suggesting that higher perceptions of physical and psychological 'caring' from providers are associated with increased use of formal supports [64]. It seems obvious, but health providers that intentionally build relationships with adolescent patients, especially over time, also improve their resilience.

An interesting outcome from this study was that many adolescents living with HIV recommended linking professionals from adolescent-friendly clinics to schools, highlighting that schools provide an untapped option to spot unreached young people living with HIV, particularly slow-progressors, and connect them to care [65]. However, consent issues obscure access in countries like South Africa with generalized HIV epidemics, where there are elevated rates of AIDS orphans, adolescents are in the custody of numerous caretakers, legal documentation of guardianship is not widespread, and guidelines around consent for HIV testing are wide-ranging, without clear direction on procedure [66].

Another noteworthy finding was the support of dynamic counseling approaches that appeal to the interests and capacities of adolescents. The effectiveness of non-verbal therapies, such as drawing, writing, drama, digital storytelling and photovoice interventions that address mental health and psychosocial issues in this population needs to be rigorously tested and controlled for. A study in Zimbabwe using body-mapping, an art therapy technique, shows value in treatment of depression in adolescents living with HIV [67]. These forms of intervention should be upscaled to lay counselors who appear well placed to utilize and deliver them with precise assessment of effectiveness [68]. It is imperative that culturally appropriate adolescent-centered interventions are developed that engage youth in a therapeutic process, acknowledge their life experiences and help them cope with their situations so that can move beyond them.

The findings from this study are limited in several ways. Firstly, all participants in this study were retained in care at the time of the interview and the factors identified by this group as significant to healthcare provision and system strengthening may differ from those of adolescents who had discontinued their care. Secondly, the adolescents in this sample were vertically infected and their views may differ significantly to horizontally infected adolescents who may not have accessed the healthcare system as intensely throughout development. However, we do believe the recommendation may be beneficial to all adolescents living with HIV regardless of mode of transmission. Thirdly, the adolescents were conveniently sampled and recruited from primary healthcare facilities in an urban setting and where adolescents communicated easily in English hence findings may not be generalizable to adolescents in other contexts or to those who are not comfortable speaking English. Fourthly, we didn't ascertain viral suppression rates in this study and successfully virally suppressed adolescents may have different impressions of healthcare needs. Lastly, our adolescents were selected from their adolescent clinics and were giving feedback on adolescent clinics. This may have led to reporting bias, however we attempted to mitigate this risk by recruiting our sample from a variety of clinics. Similarly, the healthcare staff who recruited participants may also have been influential in the accounts given by encouraging a form of social desirability bias. Notwithstanding these limitations, the views of participants in this study may be helpful in guiding services for adolescents living with HIV in South Africa.

## Conclusion

Our study adds to the literature on what adolescents feel might be of value to them in terms of HIV treatment and care within the health system, heeding the call for more participatory methods of adolescent inclusion [69]. It also gives guidance to the differentiated care model recommended for adolescent care and treatment that at its core is client centered and rights based. This care model simplifies and modifies HIV services across the cascade to both address the needs of patients more efficiently and reduce unnecessary burdens on an already over-stretched health system [60]. Concepts such as task-shifting, decentralization, community-based care all become more focal within the differentiated care model. Lay and peer counselors are a valuable, and often untapped or overlooked resource to adolescents living with HIV, both in the clinic and community, and may serve as a cornerstone of this model. In addition, serving their psychosocial and mental health needs in a responsive, caring and informed manner may strengthen adolescents' use of the healthcare system and the structure itself can be elevated to become a source of resilience for young people.

## Supporting information

**S1 File. In-depth interview guide.**
(DOCX)

## Acknowledgments

We are grateful to all participants for sharing their stories. We acknowledge the remarkable efforts of the research counselors (Thamsanqa Jabavu, Nombulelo Shezi, Linda Mazibuko, Princess Mbatha, Tebogo Moloi, Honey Nyapoli, Bafana Gxubane, Shanaaz Randeria and Relebohile Maleka) both in their capacity as counselors and for their support of this research. Thanks to Pirilani Banda for managing the administrative parts of the study. We are grateful to the Department of Health for granting access to the patients and facilities in Johannesburg, and Adv. John Peter (SC) for readily assisting to gain the court order to conduct the research.

## Author Contributions

**Conceptualization:** Nataly Woollett, Shenaaz Pahad, Vivian Black.

**Data curation:** Nataly Woollett, Shenaaz Pahad.

**Formal analysis:** Nataly Woollett, Shenaaz Pahad.

**Investigation:** Nataly Woollett, Vivian Black.

**Methodology:** Nataly Woollett.

**Project administration:** Nataly Woollett.

**Resources:** Nataly Woollett.

**Supervision:** Vivian Black.

**Validation:** Nataly Woollett.

**Writing – original draft:** Nataly Woollett.

**Writing – review & editing:** Nataly Woollett, Shenaaz Pahad, Vivian Black.

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
