## [Decision Letter · Decision Letter 0]

4 Feb 2021

PONE-D-20-34639

“We need our own clinics”: HIV infected adolescents’ recommendations for a responsive health system

PLOS ONE

Dear Dr. Woollett,

Thank you for submitting your manuscript to PLOS ONE. After careful consideration, we feel that it has merit but does not fully meet PLOS ONE’s publication criteria as it currently stands. Therefore, we invite you to submit a revised version of the manuscript that addresses the points raised during the review process.

We look forward to receiving your revised manuscript.

Kind regards,

Natella Y. Rakhmanina

Academic Editor

PLOS ONE

Journal Requirements:

2. Please include your table as part of your main manuscript and remove the individual file. Please note that supplementary tables should be uploaded as separate "supporting information" files.

3. Please ensure that you include a title page within your main document.

We do appreciate that you have a title page document uploaded as a separate file, however, as per our author guidelines (http://journals.plos.org/plosone/s/submission-guidelines#loc-title-page) we do require this to be part of the manuscript file itself and not uploaded separately.

4. Please include additional information regarding the survey or questionnaire used in the study and ensure that you have provided sufficient details that others could replicate the analyses.

For instance, if you developed a questionnaire as part of this study and it is not under a copyright more restrictive than CC-BY, please include a copy, in both the original language and English, as Supporting Information.

Additional Editor Comments:

This is an overall well written paper which provides important insight into the adolescents' needs and views on their HIV care settings and support. Minor revisions are recommended by the reviewers, and in addition authors are encouraged to consider shortening the Discussion while preserving major points to make the paper more concise.

Reviewers' comments:

Reviewer's Responses to Questions

**Comments to the Author**

1. Is the manuscript technically sound, and do the data support the conclusions?

Reviewer #1: Partly

Reviewer #2: Yes

2. Has the statistical analysis been performed appropriately and rigorously? 

Reviewer #1: Yes

Reviewer #2: N/A

3. Have the authors made all data underlying the findings in their manuscript fully available?

Reviewer #1: Yes

Reviewer #2: Yes

4. Is the manuscript presented in an intelligible fashion and written in standard English?

Reviewer #1: Yes

Reviewer #2: Yes

5. Review Comments to the Author

Reviewer #1: Thanks for a well written manuscript. The title is catchy and directs the reader to what the manuscript is about.

Introduction: Need to describe a bit more the health system as pertains South Africa, and especially in Johannesburg. Is it any different from other areas? Need to also describe more what the national guidelines state regarding transitioning of ALHIV ( Adolescents living with HIV) to adults clinic. What is the definition of adults in SA?

Methodology: The methodology is sufficiently detailed. Im concerned that the participant selection seemed a bit biased and may have affected the results. The participants seem to be ALHIV who are adherence to clinics, stable on ART, disclosed to status, retained in care for several years and attend support groups. The participants were also all fluent in English- which is another level of bias. Or was this a selection criteria at recruitment stage? So it seems likely that the results are skewed to what these "perfect" clients desire. The mention of lay workers and support for support groups is influenced by the type of participants. How would you tell of the influence the support groups of lay workers have had on those who are not stable, not been on ART for long time and have poor adherence? What of the opinion of ALHIV who are horizontally infected? who may not have been on ART for long or attended clinics for a long time? who may not have similar relations and freedom with lay counselors as those vertically infected? These new ALHIV may also have been identified in adult settings such as ANC clinics or STI clinics and may have a differing opinion on transition to adult clinics. This paper should consider opinions of both sides to be more objective. Or consider to make it clear that these are opinions of vertically infected adolescents. Would be important to describe in the results a bit more the number of participants who were screened and the proportion who were left out. What was the reason for non recruitment of these other participants attending the adolescent clinics? And how many were recruited from the once weekly clinics as opposed to the hospital that has daily adolescents clinics?

The paper describes some experiences in adult clinics. Among the participants- were there any who have been referred to adult clinics and returned to the adolescent clinics? Or were the opinions expressed based on what they have heard others say?

Additional concern on the methodology is the fact that the author did the IDI themselves. How did you ensure fidelity of the data by reducing author influence in data collection?

Results: the results are presented within context and there is visible contextualization of the quoted data. Was there consideration for participant review of the results? Was there effort to see if the participants connect with these findings?

Discussion: While part of the discussion focuses on what the findings are- including championing for support groups, need for lay workers as part of multi disciplinary teams , need to stay longer in adolescent clinics and need for widespread community education on perinatal infections, there are elements included in the discussion that were not brought out in the results section. There is discussion on issues of mental health and influence on retention and treatment outcomes. This doesn't seem to be included in the themes in the methodology or the results section. There is discussion on knowledge gaps such as on SRH and PMTCT among the ALHIV but this is not clear in the results. Furthermore, seeing that the participants were the ALHIV who have been on ART for a long time and considered in chronic care- its a surprise that they would not have knowledge on SRH and PMTCT yet these are mentioned to be discussed in the closed support groups. There is mention of Intimate partner violence among women and how it affects the ALHIV IN question. This link is not made in the results so not sure how this would be relevant. Yes, in Johannesburg violence is rampant but its not clear how this is linked to the 25 adolescents who were interviewed. The theme of Employment opportunities for ALHIV is explored much in the discussion. This is an important element that may need more unpacking.

Reviewer #2: the authors attempt to address a very important issue in pediatric HIV, with the availability of ART and decrease in mortality, the increase in the numbers of adolescents surviving and needing to transition into care is a critical issue.Understanding the adolescents' perspective is critical in order to transitioning them successfully.

There are some clarifying points/questions :

1)what was the sampling strategy, why were these 25 participants selected?

Data collection and instruments otherwise clearly described

2)how have differentiated care models been successful in other settings in overcoming the barriers identified by these adolescents?

3) What is the HIV status of these participants, viremia, CD4 cell count etc...? as participants who are successfully virally suppressed may have a different impression of health care needs etc...

6. PLOS authors have the option to publish the peer review history of their article (what does this mean?). If published, this will include your full peer review and any attached files.

Reviewer #1: No

Reviewer #2: No

---

## [Author Response · Author response to Decision Letter 0]

15 Feb 2021

13th February, 2021

Dear Dr. Heber and PLoS One Reviewers,

We thank the reviewers for their suggested edits to our manuscript entitled “We need our own clinics”: HIV infected adolescents’ recommendations for a responsive health system

 (Manuscript ID PONE-D-20-34639). 

We appreciate the time taken by the reviewers to provide detailed feedback on our manuscript and hope we have sufficiently responded to the feedback.

Sincerely,

Nataly Woollett

Editor comments:

 Thank you – we have endeavoured to complete per instruction and apologies for the oversight in the first submission.

2. Please include your table as part of your main manuscript and remove the individual file. Please note that supplementary tables should be uploaded as separate "supporting information" files.

 Apologies – the table is now included in the text.

3. Please ensure that you include a title page within your main document.

 Apologies – the title page is now included at the beginning of the document.

4. Please include additional information regarding the survey or questionnaire used in the study and ensure that you have provided sufficient details that others could replicate the analyses.

For instance, if you developed a questionnaire as part of this study and it is not under a copyright more restrictive than CC-BY, please include a copy, in both the original language and English, as Supporting Information.

The in-depth interview guide has been added as a file in supporting information.

5. This is an overall well written paper which provides important insight into the adolescents' needs and views on their HIV care settings and support. Minor revisions are recommended by the reviewers, and in addition authors are encouraged to consider shortening the Discussion while preserving major points to make the paper more concise.

Thanks for the positive feedback. We have shortened the discussion section in line with what reviewers have also recommended and hopefully this reads better.

Reviewer 1:

6. Thanks for a well written manuscript. The title is catchy and directs the reader to what the manuscript is about.

Many thanks for this positive feedback.

7. Introduction: Need to describe a bit more the health system as pertains South Africa, and especially in Johannesburg. Is it any different from other areas? Need to also describe more what the national guidelines state regarding transitioning of ALHIV ( Adolescents living with HIV) to adults clinic. What is the definition of adults in SA?

Thank you – we have added the following sentences to the introduction to address these comments:

The health system in Johannesburg is overwhelmed with large numbers of patients, often in excess of healthcare providers; leading to rapid expansion of a range of paraprofessionals being employed to address services delivery gaps (8, 9).

There are currently no national guidelines on how to transition adolescents when they reach adulthood at 18 years or before if they are stable on treatment (17).

8. Methodology: The methodology is sufficiently detailed. Im concerned that the participant selection seemed a bit biased and may have affected the results. The participants seem to be ALHIV who are adherence to clinics, stable on ART, disclosed to status, retained in care for several years and attend support groups. The participants were also all fluent in English- which is another level of bias. Or was this a selection criteria at recruitment stage? So it seems likely that the results are skewed to what these "perfect" clients desire. 

Thanks for bringing this to our attention. The participants had been retained in care for a long time, we wanted their views as they knew the health system well. But, they were not always adherent and some had histories of challenges with adherence. They were also not all retained consistently through their development, but were rather currently retained in care. We have included the following in the text to enhance clarity:

All participants reported experience of being disclosed to regarding their HIV-status, were on treatment for HIV infection, were currently retained in care, had been accessing treatment for a number of years, and participated in a clinic-based support group. Most had histories of varying levels of adherence to treatment as well as retention.

We did find most participants could communicate well in English – perhaps a factor of an urban environment. We will add this as a limitation as follows:

Thirdly, the adolescents were conveniently sampled and recruited from primary healthcare facilities in an urban setting and where adolescents communicated easily in English hence findings may not be generalizable to adolescents in other contexts or to those who are not comfortable speaking English.

9. The mention of lay workers and support for support groups is influenced by the type of participants. How would you tell of the influence the support groups of lay workers have had on those who are not stable, not been on ART for long time and have poor adherence? What of the opinion of ALHIV who are horizontally infected? who may not have been on ART for long or attended clinics for a long time? who may not have similar relations and freedom with lay counselors as those vertically infected? These new ALHIV may also have been identified in adult settings such as ANC clinics or STI clinics and may have a differing opinion on transition to adult clinics. This paper should consider opinions of both sides to be more objective. Or consider to make it clear that these are opinions of vertically infected adolescents. 

Thanks for raising this point. This qualitative study was nested in a larger quantitative study that administered a questionnaire to 343 adolescents at 5 HIV clinics. The clinics were in a variety of areas in Johannesburg and regular standard of care was given at all except 1 that was a flagship clinic with an enhanced standard of care. The participants in the qualitative study were recruited equally from the 5 sites so were not a biased sample from the flagship clinic. To our surprise, of the 343 adolescents recruited for the quantitative study, only 4 participants were known to be horizontally infected and they were from the same regular standard of care clinic. As such, the majority of the sample was vertically infected and all those for the qualitative study were vertically infected too. We didn’t recruit in ANC or STI clinics and agree, in those environments, we could have harnessed a more diverse patient population.

We draw the reviewers attention to this sentence in the methods section that does state the sample was vertically infected:

Participants were infected at birth or early childhood and had lived with HIV as long as they remembered.

And also this sentence in the limitations section:

Secondly, the adolescents in this sample were vertically infected and their views may differ significantly to horizontally infected adolescents who may not have accessed the healthcare system as intensely throughout development.

10. Would be important to describe in the results a bit more the number of participants who were screened and the proportion who were left out. What was the reason for non recruitment of these other participants attending the adolescent clinics? And how many were recruited from the once weekly clinics as opposed to the hospital that has daily adolescents clinics?

Thank you for the recommendation – we have added the following sentence to the methods section:

The first five participants that met inclusion criteria at each of the five sites were invited to participate in the qualitative interviews. No participants refused and we felt we reached saturation with 25 participants in total.

11. The paper describes some experiences in adult clinics. Among the participants- were there any who have been referred to adult clinics and returned to the adolescent clinics? Or were the opinions expressed based on what they have heard others say?

Yes, there were a couple of participants who had been referred and came back to the adolescent clinic. Kagiso, Bafense and Sithembile experienced this. To increase clarity, we have altered the sentence to read as follows:

Many stable adolescent patients had been down referred to adult clinics and soon returned to the adolescent clinics, Kagiso and Bafense below remembered this experience.

12. Additional concern on the methodology is the fact that the author did the IDI themselves. How did you ensure fidelity of the data by reducing author influence in data collection?

Thanks, however, the purpose was not to be wholly objective, since every qualitative researcher approaches data collection with explicit or implicit understandings of the phenomenon at hand (Maxwell, 1992), but rather interpret the findings in light of the participant voices themselves. For this reason we draw heavily on verbatim participant quotes to illustrate key points. In addition, the analysis of the data was undertaken by multiple researchers to ensure consensus and guarantee research findings. This is from the data analysis section:

To ensure intercoder consensus, fine codes were developed by three researchers experienced in qualitative data analysis by printing out a full set of excerpts (from each data set) associated with each code for each transcript and identifying sub-themes arising from the data. Two researchers applied the thematic codes to every transcript. The findings were critiqued and discussed within the group to guarantee research results.

13. Results: the results are presented within context and there is visible contextualization of the quoted data. Was there consideration for participant review of the results? Was there effort to see if the participants connect with these findings?

We didn’t go back to participants with the findings unfortunately, however, our adolescent community advisory group did engage with the findings and seemed to relate to them. All ACAB members were adolescents that were engaged in care in the public health system in Johannesburg. The following sentence has been added at the end of the ‘Participants and Procedures’ section:

The ACAB gave advice on methods and were also engaged with the findings of the study.

14. Discussion: While part of the discussion focuses on what the findings are- including championing for support groups, need for lay workers as part of multi disciplinary teams , need to stay longer in adolescent clinics and need for widespread community education on perinatal infections, there are elements included in the discussion that were not brought out in the results section. There is discussion on issues of mental health and influence on retention and treatment outcomes. This doesn't seem to be included in the themes in the methodology or the results section. 

 We bring the reviewers attention to this part of the results section:

Support groups also facilitated mental health gains such as hope, optimism and relief of burdensome emotions.

They will have to support us in what we are doing because we will help this virus to get down and down because if they are not supporting us, this virus will busy getting up and higher because we are losing a lot of youth because they are scared to be HIV, but as we are, we are not scared cause they mustn’t lose hope, there is hope out there, we can help them handle person with pain – Sithembile, male, 18yrs

Discuss everything that you want to discuss about you take out all the pain inside you, there by the group – Tebogo, female, 18yrs

And this section of the results as well:

 Participants seemed to rely heavily on counselors for information about HIV and their treatment, but also sought guidance and emotional support from them.

We need to put more counselors in clinics to support us, maybe telling us what is going on about our lives, what should we do if there is time that’s come difficult maybe – Nomphumelelo, female, 17yrs

15. There is discussion on knowledge gaps such as on SRH and PMTCT among the ALHIV but this is not clear in the results. Furthermore, seeing that the participants were the ALHIV who have been on ART for a long time and considered in chronic care- its a surprise that they would not have knowledge on SRH and PMTCT yet these are mentioned to be discussed in the closed support groups. 

 Yes, this was a surprising findings for the research team too. SRH and PMTCT had been spoken about in multiple contexts with these young patients by multiple adults, at clinic level, within groups and presumably at school as well. However, either the concepts were not explained clearly and articulately or they were not understood well for these twenty five participants. Our research team discussed why this might be the case at length and concluded perhaps with these more ‘technical’ issues in HIV treatment and care, the words and concepts are stated but the particularities are not discussed at length at all or how prevention methods really prevent disease transmission for individuals. Perhaps adults who manage vertically infected adolescents assume they know this material. All the participants had heard of PMTCT and would say it was to prevent transmission but they didn’t actually understand how it prevented transmission. In situations of teaching when young people are expected to listen, perhaps there isn’t much agency to ask adults directly what they mean and perhaps adults don’t necessarily have the technical detail and comprehensive understanding to explain well. This following sentence has been added to the findings to aid in clarity:

 In fact, the interviewer explained PMTCT at length to all participants as there was not comprehensive understanding of this prevention method in the sample.

16. There is mention of Intimate partner violence among women and how it affects the ALHIV IN question. This link is not made in the results so not sure how this would be relevant. Yes, in Johannesburg violence is rampant but its not clear how this is linked to the 25 adolescents who were interviewed. 

 In our larger study that these participants were recruited from, there was high exposure and experience of multiple forms of violence. We have now added a reference to that study and included a sentence that makes the link as follows:

 Many vertically infected children and adolescents live in communities characterized by poverty and exposure to violence (37) as was the case in our sample (31). 

17. The theme of Employment opportunities for ALHIV is explored much in the discussion. This is an important element that may need more unpacking.

 Thank you, we agree. Unfortunately, we are constrained by the word limitation and other reviewers recommendations to shorten the discussion section. We trust that the paragraph in the discussion section will suffice on this topic.

Reviewer 2: 

18. The authors attempt to address a very important issue in pediatric HIV, with the availability of ART and decrease in mortality, the increase in the numbers of adolescents surviving and needing to transition into care is a critical issue. Understanding the adolescents' perspective is critical in order to transitioning them successfully.

 We agree and value your comment thank you.

19. What was the sampling strategy, why were these 25 participants selected?

 We have added these sentences to the methods section to explain further:

 The first five participants that met inclusion criteria at each of the five sites were invited to participate in the qualitative interviews. No participants refused and we felt we reached saturation with 25 participants in total.

20. Data collection and instruments otherwise clearly described.

 Great thank you.

21. How have differentiated care models been successful in other settings in overcoming the barriers identified by these adolescents?

 We have identified a couple of areas, such as community-based adherence clubs, support groups (otherwise known as ‘Teen Clubs’) and mental health counselling and make recommendation of them in the discussion:

 Community-based adherence clubs that have shown positive outcomes for adult patients (58) may be equally effective for adolescents, especially if there is a support group component to implementation and if the groups are adolescent only (29).

 There is growing evidence that clinic-based support groups help retain HIV infected adolescents in care and if managed well, can be taken to scale (14, 32).

Those (support groups) that intentionally address some of the mental health difficulties of HIV infected adolescents and that are facilitated by staff grounded in understanding adolescent development and mental health, could mitigate risk, improve health system use and treatment adherence.

22. What is the HIV status of these participants, viremia, CD4 cell count etc...? as participants who are successfully virally suppressed may have a different impression of health care needs etc.

 This is a valuable question and regretably not one that we investigated. We didn’t verify viral suppression rates or review medical records in this study unfortunately. We will add this as a limitation as follows:

 Fourthly, we didn’t ascertain viral suppression rates in this study and successfully virally suppressed adolescents may have different impressions of healthcare needs.

Reference:

Maxwell J. Understanding and validity in qualitative research. Harvard educational review. 1992 Sep 1;62(3):279-301.

---

## [Decision Letter · Decision Letter 1]

19 Apr 2021

PONE-D-20-34639R1

“We need our own clinics”: HIV infected adolescents’ recommendations for a responsive health system

PLOS ONE

Dear Dr. Woollett,

Thank you for submitting your manuscript to PLOS ONE. After careful consideration, we feel that it has merit but does not fully meet PLOS ONE’s publication criteria as it currently stands. Therefore, we invite you to submit a revised version of the manuscript that addresses the points raised during the review process.

ACADEMIC EDITOR: Thank you for your submission of this important manuscript. We feel that this manuscript has merit but requires minor revision before final acceptance. Please address Reviewer #3 comments, in particular the use of person-first language throughout, description of coding conflicts, and the formatting of Table 1. Also respond to Review #3's comment about higher order analysis which would strengthen the manuscript but would not be required.

We look forward to receiving your revised manuscript.

Kind regards,

Brian C. Zanoni, MD

Academic Editor

PLOS ONE

Journal Requirements:

Additional Editor Comments (if provided):

Please address the comments by reviewer 3 in your revised manuscript.

Reviewers' comments:

Reviewer's Responses to Questions

**Comments to the Author**

1. If the authors have adequately addressed your comments raised in a previous round of review and you feel that this manuscript is now acceptable for publication, you may indicate that here to bypass the “Comments to the Author” section, enter your conflict of interest statement in the “Confidential to Editor” section, and submit your "Accept" recommendation.

Reviewer #1: All comments have been addressed

Reviewer #3: (No Response)

2. Is the manuscript technically sound, and do the data support the conclusions?

Reviewer #1: Yes

Reviewer #3: Yes

3. Has the statistical analysis been performed appropriately and rigorously? 

Reviewer #1: N/A

Reviewer #3: Yes

4. Have the authors made all data underlying the findings in their manuscript fully available?

Reviewer #1: Yes

Reviewer #3: Yes

5. Is the manuscript presented in an intelligible fashion and written in standard English?

Reviewer #1: Yes

Reviewer #3: Yes

6. Review Comments to the Author

Reviewer #1: (No Response)

Reviewer #3: This is an important study given the increasing numbers of adolescents living with HIV surviving into adulthood and eventual need to transition to adult-oriented care settings. The authors attempt to address this critical time by examining the attitudes and experiences of perinatally infected adolescents living with HIV regarding transition from pediatric to adult care in South Africa, including their recommendations for a successful transition. The study provided insights from adolescents about this critical time through in-depth interviews, including the importance of clinic-based support groups, the value of lay counselors in providing education, counseling, and support to adolescents, increasing widespread education of different modes of HIV transition (especially vertical transmission), and linking the clinic to the community.

The authors have adequately responded to the previous reviewers’ comments; This is my first review of this revised manuscript. Overall, the methods, interpretation, and communication of the findings are excellent. The major strengths of this manuscript are the focus on a population (perinatally-infected adolescents living with HIV in South Africa) that has high rates of morbidity and mortality at each point of the continuum of care, especially healthcare transition, as well as a richness of quotations from the participants to support your discussion. However, a few additional points to consider, including the use of identity-first vs person-first language, unclear methodology with regards to intercoder agreement, incorrect formatting of table 1, and lack of synthesis of data to broader analytical themes.

1. I am concerned with the authors use of identity-first language in the title and throughout the manuscript. The authors should use person-first language in the title and throughout the manuscript. For example: “HIV infected adolescents” can be changed to “adolescents living with HIV;” and “HIV infected women” can be changed to “women living with HIV.”

2. Additional concern with the methodology is regarding the intercoder reliability; specifically how were discrepancies between independent coders resolved?

3. Within the results section, table 1 is formatted incorrectly with the table split with text dividing the table. Authors should fix the formatting for table 1.

4. An additional concern with the results section is the authors dependence on quotations and their interpretation of these quotations for development of descriptive themes. The authors could consider synthesizing results to a higher order to generate new analytical themes or explanations.

7. PLOS authors have the option to publish the peer review history of their article (what does this mean?). If published, this will include your full peer review and any attached files.

Reviewer #1: No

Reviewer #3: No

---

## [Author Response · Author response to Decision Letter 1]

18 May 2021

Reviewer #3: This is an important study given the increasing numbers of adolescents living with HIV surviving into adulthood and eventual need to transition to adult-oriented care settings. The authors attempt to address this critical time by examining the attitudes and experiences of perinatally infected adolescents living with HIV regarding transition from pediatric to adult care in South Africa, including their recommendations for a successful transition. The study provided insights from adolescents about this critical time through in-depth interviews, including the importance of clinic-based support groups, the value of lay counselors in providing education, counseling, and support to adolescents, increasing widespread education of different modes of HIV transition (especially vertical transmission), and linking the clinic to the community.

The authors have adequately responded to the previous reviewers’ comments; This is my first review of this revised manuscript. Overall, the methods, interpretation, and communication of the findings are excellent. The major strengths of this manuscript are the focus on a population (perinatally-infected adolescents living with HIV in South Africa) that has high rates of morbidity and mortality at each point of the continuum of care, especially healthcare transition, as well as a richness of quotations from the participants to support your discussion. 

 Thank you Reviewer 3 for that positive feedback.

However, a few additional points to consider, including the use of identity-first vs person-first language, unclear methodology with regards to intercoder agreement, incorrect formatting of table 1, and lack of synthesis of data to broader analytical themes.

1. I am concerned with the authors use of identity-first language in the title and throughout the manuscript. The authors should use person-first language in the title and throughout the manuscript. For example: “HIV infected adolescents” can be changed to “adolescents living with HIV;” and “HIV infected women” can be changed to “women living with HIV.”

Thank you for bringing this important point to our attention. Please note that person-first language has now been incorporated throughout the text.

2. Additional concern with the methodology is regarding the intercoder reliability; specifically how were discrepancies between independent coders resolved?

A group of 3 researchers where part of the analysis team and we double coded every transcript. We met weekly to discuss the coding, resolve any conflicts and learn from the data in an iterative way. We were not aiming for perfect alignment in terms of intercoder reliability but rather consensus in terms of the meaning of codes and we felt we were able to reach this successfully through this process. We have added the following to improve clarity:

To ensure intercoder consensus, fine codes were developed by three researchers experienced in qualitative data analysis by printing out a full set of excerpts (from each data set) associated with each code for each transcript and identifying sub-themes arising from the data. Two researchers applied the thematic codes to every transcript. Each week during the double coding, the findings were critiqued and discussed within the group to guarantee research results. During analysis, we found that data reached a point of saturation, which suggests the sample size was sufficient.

3. Within the results section, table 1 is formatted incorrectly with the table split with text dividing the table. Authors should fix the formatting for table 1.

Apologies, the table has been realigned now. Thanks.

4. An additional concern with the results section is the authors dependence on quotations and their interpretation of these quotations for development of descriptive themes. The authors could consider synthesizing results to a higher order to generate new analytical themes or explanations.

 Although higher order analysis may strengthen this manuscript, we do believe we were able to use a grounded theory approach to analysis effectively. We were interested in analyzing the data and deriving meaning from it in the first instance. We feel through this method we were able to meaningfully construct and describe the views and lives of our participants, adding to the literature around their recommendations for their healthcare and bringing their voices more demonstrably to that conversation.

Please also note that reference to Table 1 has been made within the text as required.

---

## [Decision Letter · Decision Letter 2]

17 Jun 2021

“We need our own clinics”: Adolescents living with HIV recommendations for a responsive health system

PONE-D-20-34639R2

Dear Dr. Woollett,

We’re pleased to inform you that your manuscript has been judged scientifically suitable for publication and will be formally accepted for publication once it meets all outstanding technical requirements.

Kind regards,

Brian C. Zanoni, MD

Academic Editor

PLOS ONE

Additional Editor Comments (optional):

Reviewers' comments:

Reviewer's Responses to Questions

**Comments to the Author**

1. If the authors have adequately addressed your comments raised in a previous round of review and you feel that this manuscript is now acceptable for publication, you may indicate that here to bypass the “Comments to the Author” section, enter your conflict of interest statement in the “Confidential to Editor” section, and submit your "Accept" recommendation.

Reviewer #1: All comments have been addressed

Reviewer #3: All comments have been addressed

2. Is the manuscript technically sound, and do the data support the conclusions?

Reviewer #1: Yes

Reviewer #3: Yes

3. Has the statistical analysis been performed appropriately and rigorously? 

Reviewer #1: Yes

Reviewer #3: Yes

4. Have the authors made all data underlying the findings in their manuscript fully available?

Reviewer #1: Yes

Reviewer #3: Yes

5. Is the manuscript presented in an intelligible fashion and written in standard English?

Reviewer #1: Yes

Reviewer #3: Yes

6. Review Comments to the Author

Reviewer #1: The authors have responded satisfactorily to the reviewer comments. There was concern on the methodology and the analysis and these have been addressed.

Reviewer #3: The authors have provided a nicely detailed and thorough response to the comments from the previous review and have addressed my concerns. The study findings are important and will be of interest to a broad audience.

7. PLOS authors have the option to publish the peer review history of their article (what does this mean?). If published, this will include your full peer review and any attached files.

Reviewer #1: No

Reviewer #3: No

---

## [Editor Report · Acceptance letter]

23 Jun 2021

PONE-D-20-34639R2 

“We need our own clinics”: Adolescents’ living with HIV recommendations for a responsive health system 

Dear Dr. Woollett:

I'm pleased to inform you that your manuscript has been deemed suitable for publication in PLOS ONE. Congratulations! Your manuscript is now with our production department. 

Kind regards, 

on behalf of

Dr. Brian C. Zanoni 

Academic Editor

PLOS ONE